# Mimicking Protein Kinase C Phosphorylation Inhibits Arc/Arg3.1 Palmitoylation and Its Interaction with Nucleic Acids

**DOI:** 10.3390/ijms25020780

**Published:** 2024-01-08

**Authors:** Barbara Barylko, Clinton A. Taylor, Jason Wang, Svetlana Earnest, Steve Stippec, Derk D. Binns, Chad A. Brautigam, David M. Jameson, George N. DeMartino, Melanie H. Cobb, Joseph P. Albanesi

**Affiliations:** 1Department of Pharmacology, U.T. Southwestern Medical Center, 6001 Forest Park, Dallas, TX 75390, USA; bbarylko@utsouthwestern.edu (B.B.); clinton.taylor@utsouthwestern.edu (C.A.T.4th); derk.binns@utsouthwestern.edu (D.D.B.); melanie.cobb@utsouthwestern.edu (M.H.C.); 2Department of Physiology, U.T. Southwestern Medical Center, 6001 Forest Park, Dallas, TX 75390, USA; jason.wang2@utsouthwestern.edu (J.W.); george.demartino@utsouthwestern.edu (G.N.D.); 3Department of Biophysics, U.T. Southwestern Medical Center, 6001 Forest Park, Dallas, TX 75390, USA; chad.brautigam@utsouthwestern.edu; 4Department of Cell and Molecular Biology, John A. Burns School of Medicine, University of Hawaii, Honolulu, HI 96844, USA; djameson@hawaii.edu

**Keywords:** Arc/Arg3.1, phosphorylation, protein kinase C, nucleic acid binding, oligomerization, palmitoylation, post-translational modifications, neuronal activity

## Abstract

Activity-regulated cytoskeleton-associated protein (Arc) plays essential roles in diverse forms of synaptic plasticity, including long-term potentiation (LTP), long-term depression (LTD), and homeostatic plasticity. In addition, it assembles into virus-like particles that may deliver mRNAs and/or other cargo between neurons and neighboring cells. Considering this broad range of activities, it is not surprising that Arc is subject to regulation by multiple types of post-translational modification, including phosphorylation, palmitoylation, SUMOylation, ubiquitylation, and acetylation. Here we explore the potential regulatory role of Arc phosphorylation by protein kinase C (PKC), which occurs on serines 84 and 90 within an α-helical segment in the N-terminal domain. To mimic the effect of PKC phosphorylation, we mutated the two serines to negatively charged glutamic acid. A consequence of introducing these phosphomimetic mutations is the almost complete inhibition of Arc palmitoylation, which occurs on nearby cysteines and contributes to synaptic weakening. The mutations also inhibit the binding of nucleic acids and destabilize high-order Arc oligomers. Thus, PKC phosphorylation of Arc may limit the full expression of LTD and may suppress the interneuronal transport of mRNAs.

## 1. Introduction

Activity-dependent cytoskeletal-associated protein (Arc) [1], also known as Arg3.1 [2], is a critical regulator of various forms of synaptic plasticity, including long-term potentiation (LTP), long-term depression (LTD), and homeostatic scaling [3]. The critical role of Arc in synaptic plasticity is highlighted by findings of abnormal Arc expression and function in neurological disorders such as Alzheimer’s disease, Parkinson’s disease, epilepsy, and schizophrenia [3,4]. The Arc gene is rapidly induced in organisms in response to behavioral events and in cultured cells in response to electrical or chemical stimuli [5,6,7]. Arc is expressed predominantly in the nucleus, dendrites, and postsynaptic density (PSD) of cortical and hippocampal glutamatergic neurons. A pool of Arc mRNA traffics through the dendritic arbor in association with a ribonucleoprotein complex and is locally translated upon postsynaptic stimulation. Within dendritic spines, Arc is believed to promote synaptic weakening (LTD) by facilitating endocytosis of AMPA receptors [8,9,10,11] and synaptic strengthening by reorganization of the actin cytoskeleton [12,13]. More recently, a second potential mechanism of Arc mRNA translocation was revealed. The three-dimensional structure of a C-terminal fragment (residues 217–362) was solved and was found to be similar to that of the HIV Gag capsid domain [14]. Findings that the Arc gene has a retroviral origin [15] and bears a structural similarity to the HIV-1 capsid domain led to the remarkable discovery of a different mechanism of Arc mRNA translocation, namely, the encapsulation into virus-like Arc particles that are released as extracellular vesicles and internalized by neighboring cells [16,17].

Considering the diverse neuronal functions governed by Arc, it is not surprising that its activities are regulated by numerous post-translational modifications [18], including ubiquitylation, acetylation, SUMOylation, and palmitoylation (Figure 1). Ubiquitylation [19] and acetylation [20] control Arc expression levels by promoting or suppressing proteasomal degradation, respectively. SUMOylation, which regulates Arc function in synaptic scaling [21], strengthens Arc’s association with drebrin A, a protein that promotes actin assembly in dendritic spines [22]. Palmitoylation of Arc, which was first reported by our laboratory, is required for maximal synaptic weakening in response to LTD protocols [23]. Arc has also been reported to undergo phosphorylation by several kinases, including glycogen synthase kinase (GSK)3α and GSK3β, which regulate its proteasomal degradation [24], extracellular signal-regulated kinase (ERK), which regulates its subcellular localization [25], Ca^2+^/calmodulin-dependent protein kinase IIα (CaMKIIα), which regulates its oligomerization [26], and TRAF2 and NCK-interacting protein kinase (TNIK), which regulates both its subcellular distribution and self-assembly [27]. Although protein kinase C (PKC) has long been implicated in both LTP and LTD [28], the function and regulation of PKC-mediated Arc phosphorylation have not been examined. In this study, we have identified serines 84 and 90 as the major in vitro PKC-phosphorylated residues. These residues reside within the second α-helical segment (helix H2) in the N-terminal domain of Arc. To mimic the effect of phosphorylation, we mutated the two serines to negatively charged glutamic acid. These “phosphomimetic” mutations resulted in a reduced propensity of Arc to undergo palmitoylation, reduced binding to nucleic acids, and reduced stability of high-order oligomers. The latter two characteristics were also displayed by the deletion of H2 and of segments within H2. Taken together, these results suggest that PKC phosphorylation within the H2 helix suppresses palmitoylation-dependent aspects of LTD and intercellular transport of RNA by virus-like Arc particles. 

## 2. Results

### 2.1. Identification of PKC Phosphorylation Sites in Arc

To identify regions in the Arc molecule that contain the major PKC phosphorylation sites, we first mapped its approximate domain boundaries by limited proteolysis. As shown by Myrum et al. [29] and secondary structure predictions, such as PSIPRED [30] (Figure 2A), Arc can be divided roughly into two domains connected by a long intrinsically disordered segment, which is hypersensitive to proteolysis. Bacterially expressed His_6_-Arc was subjected to limited papain digestion, and the most stable fragments were identified using mass spectrometry and N-terminal sequencing. Three major fragments of approximately 33, 30, and 24 kDa (by SDS-PAGE) were obtained (Figure 2B), with all three originating from the C-terminal portion of Arc and initiating, respectively, at residues Val^152^, Ile^179^, and Ser^195^. Based on these results, we attempted to express and purify glutathione-S-transferase (GST)-tagged N-terminal domain (NTD; residues 1–151) and C-terminal domain (CTD; residues 152–396) fragments. GST-Arc^1–151^ did not express well, so we expressed full-length GST-Arc with two tobacco etch virus (TEV) protease sites, one after GST (to remove GST) and the other after residue 151 to obtain the N- and C-terminal fragments. Although Arc^152–396^ was completely soluble, Arc^1–151^ fell out of solution immediately upon proteolysis (Figure 2C). This finding was consistent with that of Hallin et al. [31], who reported that C-terminal constructs of Arc are soluble, even at concentrations greater than 10 mg/mL, whereas Arc^1–130^ is insoluble.

Arc^152–396^ has a calculated molecular weight of ~28 kDa but an apparent molecular weight by SDS gel electrophoresis of ~36 kDa. The anomalous electrophoretic migration of this fragment is likely due to its low calculated isoelectric point of 4.42. Sedimentation velocity analysis (Figure 2D) revealed that Arc^152–396^ is a monodisperse protein with a sedimentation coefficient of approximately 1.9 S, a molecular weight of ~29,000, and a frictional ratio of ~1.70–1.85, considerably higher than the f/f_0_ range of about 1.15–1.25 characteristic of globular proteins [32]. A high frictional ratio is indicative of protein asymmetry and/or extensive intrinsic disorder and flexibility. Arc^152–396^ has a calculated molecular mass of ~28 kDa, confirming prior reports that the Arc CTD is almost exclusively monomeric in solution [31]. This finding was supported by chemical cross-linking analysis using glutaraldehyde (Figure 2E).

**Figure 2 ijms-25-00780-f002:**
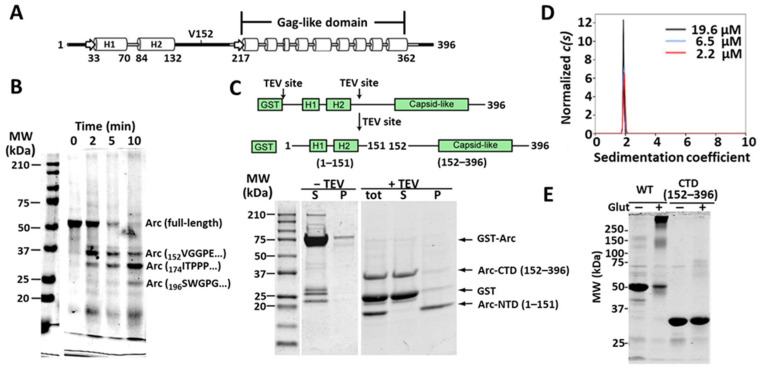
Identification of N- and C-terminal domains of Arc. (**A**) Prediction of secondary structure of mouse Arc using the PSIPRED program [30]. Arrows: β-strands; cylinders: α-helices; black lines: intrinsically disordered segments, as predicted by DisEMBL [33]. (**B**) Proteolytic cleavage of His_6_-Arc by papain at 26 °C and an enzyme:Arc ratio of 1:500 for indicated times. N-terminal sequences of selected fragments, obtained by Edman degradation, are indicated. (**C**) Preparation of GST-Arc with two TEV cleavage sites before and after incubation with TEV. Supernatants (S) and pellets (P) were obtained after centrifugation at 230,000× *g* for 15 min. tot: total digest. GST-Arc was cleaved at a TEV:Arc ratio of 1:100 for 4 h at 4 °C. In panels B and C, samples were run on 12% SDS-polyacrylamide gels and stained with Coomassie blue. (**D**) Sedimentation velocity size distributions of three concentrations of Arc^152–396^. The c(s) distributions have been normalized by the areas under the respective curves. Concentrations are shown in the inset. (**E**) Gluteraldehyde cross-linking of Arc^WT^ (10 µM) and Arc^152–396^ (18 µM). Samples were cross-linked with 0.05% glutaraldehyde for 10 min. Samples were subjected to SDS-PAGE on a 10% gel, which was then stained with Coomassie blue.

Using the double-TEV GST-Arc construct, we determined that residues phosphorylated by PKC in vitro are concentrated in Arc^NTD^, whereas those phosphorylated by the MAP kinase ERK2 are located almost exclusively within Arc^CTD^, as reported previously [25] (Figure 3A). Arc contains four serine residues (S84, S90, S260, and S390) predicted to undergo phosphorylation by PKC [2]. Two of these putative PKC sites, S84 and S90, are located within the second α-helical segment of Arc^NTD^ (residues 84–132, which we designate as “H2” in Figure 1). Phosphorylation of S84 was detected in a high-throughput screen (PhosphoSite) [34] but has not been further investigated. Deletion of H2 diminished PKCα-mediated phosphorylation of Arc by approximately 6-fold (Figure 3B,C). We confirmed by mutational analysis that serines 84 and 90 account for at least 80% of the PKC phosphorylation of segment H2 (Figure 3D). Figure 3E shows the structure of residues 29–137, encompassing helices H1 and H2, as predicted by AlphaFold [35,36]. The PKC phosphorylation sites are near the hinge between the two helices and in close proximity to the palmitoylation motif. 

### 2.2. Introduction of PKC Phosphomimetic Mutations Does Not Prevent High-Order Oligomerization of Arc

At physiologic pH and ionic strength, Arc self-associates in vitro into multiple oligomeric species with diameters ranging from about 4 nm (monomer) to 20–35 nm (30–40-mers) [16,17,29,37]. The physiological significance of Arc self-assembly has not been conclusively established, but its potential importance was highlighted by the aforementioned demonstration that Arc forms virus-like particles that may encapsulate mRNA for delivery into the extra-neuronal space [16,17]. To test whether the introduction of negative charges at PKC-phosphorylated sites affects Arc self-association, we compared the hydrodynamic radii of Arc^WT^ and Arc^S84,90E^ at ~20 µM concentration using dynamic light scattering (DLS). As shown in Figure 4A,B, Arc^WT^ and Arc^S84,90E^ had hydrodynamic radii (R_H_) of ~22 nm and 17.5 nm, respectively. These values are consistent with those obtained previously by DLS and electron microscopy and indicate that phosphomimetic mutations do not inhibit the formation of high-order Arc oligomers. However, we cannot use these R_H_ values to estimate precise oligomeric states, as Arc is ~40% intrinsically disordered and, hence, is unlikely to assume an ideal globular morphology.

### 2.3. Effect of Phosphomimetic Mutations on Palmitoylation 

Palmitoylation is the reversible attachment of 16-carbon palmitoyl chains, usually to cysteines, catalyzed by specific protein acyltransferases (PATs). We reported that Arc undergoes palmitoylation in neurons and localized the palmitoylated cysteines to a motif (_94_CLCRC_98_) within the N-terminal region of helix H2 [23]. Expression of a non-palmitoylatable mutant in Arc knockout mice failed to rescue Arc-dependent mechanisms of synaptic depression. Although a relatively small percentage of Arc is palmitoylated, the specific palmitoyl-Arc pool may play important roles in the neuron. For example, <1% of PICK1 is palmitoylated in neurons, yet palmitoylated PICK1 is required for the induction of LTD in the cerebellum [39]. Because the palmitoylation motif is in close proximity to the PKC phosphorylation sites, we tested whether the introduction of the S-to-E mutations of residues Ser84 and Ser90 influenced Arc palmitoylation. To do so, we used the acyl-resin-assisted capture (acyl-RAC) method, whereby palmitoylated proteins are first treated with methyl methanethiosulfonate to block free thiols, then incubated with hydroxylamine (NH_2_OH) to expose hitherto palmitoylated sulfhydryls, captured on thiopropyl-Sepharose resin, and identified by immunoblotting [40]. Parallel samples are incubated with NaCl instead of NH_2_OH to control for false positives. As shown in Figure 5A, palmitoylation (i.e., capture after NH_2_OH treatment) was reduced almost to the same levels by mutation of the palmitoylation sites (C94,96,98S), by double mutation of the PKC phosphorylation sites (S84,90E), or by triple mutation of the PKC sites and of the only other phosphorylatable residue in H2, Ser132 (S84,90,132E). We note that Ser 132 is predicted to be a substrate for CaMKII. Results are quantified in Figure 5B. Because the phosphomimetic mutations decrease palmitoylation to essentially the same levels as the mutation of the palmitoylated cysteines, we conclude that PKC phosphorylation abrogates Arc palmitoylation.

### 2.4. Effect of Phosphomimetic Mutations on Nucleic Acid Binding 

Arc interacts with nucleic acids, accounting for the relatively high ratios of absorbance at 260 and 280 nm (A_260/280_) of about 1.0 observed for purified, bacterially expressed Arc [16]. The stripping Arc of its bound RNA reduces its A_260/280_ ratio to about 0.7 and prevents its assembly into a relatively homogeneous population of ~32 nm particles that resemble viral capsids in cryo-electron micrographs [16]. Moreover, the addition of either an irrelevant mRNA (that of GFP) or Arc mRNA to purified Arc protein was found to increase its oligomerization [41]. We previously reported that precipitation of bacterially expressed Arc with 35% ammonium sulfate (AS) strips Arc of its bound (presumably bacterial) mRNA, reducing its A_260/280_ from ~1.0 to ~0.6, as expected for a nucleic acid-free protein [42]. In the spectrum presented in Figure 6A (top panel), Arc^WT^ purified without AS precipitation displays an A_260/280_ of ~0.9. In contrast, the A_260/280_ of Arc^S84,90E^ purified without AS precipitation is ~0.55 (Figure 6A, bottom panel), indicating that phosphomimetic PKC mutations inhibit nucleic acid binding. As expected, AS treatment diminished the A_260/280_ ratio of Arc^WT^ to ~0.55 (Figure 6B, top panel) but had no effect on the A_260/280_ of Arc^S84,90E^ (Figure 6B, bottom panel). Consistent with these results, the supernatant obtained after AS-mediated stripping of nucleic acids from Arc^WT^ had an A_260/280_ of ~2.0, indicative of pure RNA (Figure 6C, top panel), whereas almost no signal was evident in the supernatant obtained after AS precipitation of Arc^S84,90E^ (Figure 6C, bottom panel). Thus, PKC phosphorylation appears to inhibit the interaction of Arc with mRNA. 

### 2.5. Effect of Phosphomimetic Mutations on the Elution of Arc from Gel Filtration Columns 

Arc elutes from size-exclusion chromatography (SEC) columns in several distinct but overlapping peaks, consistent with the presence of at least three low-order oligomeric species (Stokes’ radii (R_S_) of ~7 nm, ~6.1 nm, and ~4.4 nm) and one high-order (greater than 20-mer) species eluting near the void volume (V_o_) [37,42]. A typical elution profile from a Superdex 200 column (exclusion limit = 1.3 × 10^6^ Da) is reproduced in Figure 7A. As we previously reported [42], the removal of bound nucleic acids by 35% AS precipitation eliminated the high-order oligomeric peak (Figure 7B). We note that the DLS analysis shown in Figure 4A was performed with AS-treated (and, hence, nucleotide-free) Arc^WT^ at concentrations similar to those used in our SEC measurements. The presence of high-order oligomers in the DLS scans, but not in the gel filtration profiles, of AS-treated Arc^WT^ suggests that high-order oligomers dissociate into lower-order species as they elute through the ~100 mL gel filtration columns. Nucleic acids apparently stabilize high-order Arc oligomers, suppressing their complete dissociation into low-order species during gel filtration. The high-order oligomeric peak is not evident in gel filtration profiles of AS-untreated Arc^S84,90E^ (Figure 7C), consistent with the absence of bound nucleotides.

The H2 domain was previously implicated in nucleic acid binding by the observation that mutation of residues 113–119 inhibits the enhancement of Arc self-assembly by exogenous mRNAs [41]. We sought to further localize nucleic acid binding determinants in H2 by measuring A_260/280_ of a set of Arc deletion mutants lacking either the entire H2 domain (Arc^ΔH2^) or only residues 91–100 (the endophilin binding motif [8]), 94–127 (which contains an Arc dimerization determinant [41]), or 108–124 (which folds into an amphipathic helix). We also measured A_260/280_ of the CTD (residues 152–396) and of a deletion mutant lacking helix H1 (residues 33–70, which was originally termed the “coiled-coil domain”). All proteins were prepared with or without AS precipitation to test for the stripping of bound nucleic acids. As shown in Table 1, deletion of H2 or any segment within H2 eliminated nucleic acid binding, even in the absence of AS treatment. Likewise, no nucleic acid binding to the CTD was observed. In contrast, Arc^ΔH1^ displayed similar nucleic acid binding to Arc^WT^ and retained significant binding even upon AS precipitation. Thus, it appears that the presence of helix H1, which is in close contact with helix H2 according to AlphaFold prediction (Figure 3E), suppresses the H2–mRNA interaction, whereas its removal strengthens the interaction, rendering it less sensitive to AS-mediated stripping. 

To test whether the absence of bound nucleic acids destabilizes high-order Arc oligomers, we subjected each of the deletion mutants to gel filtration chromatography. Figure 8A,B shows that the removal of helix H2 or segments within H2 results in the loss of the peak eluting near the void volume of Superdex 200 columns, despite samples being prepared without AS treatment. In contrast, a large pool of AS-treated or untreated Arc^ΔH1^ elutes near the void volume, supporting the possibility that helix H1 is an auto-inhibitor of nucleic acid binding. 

## 3. Discussion

This report introduces potential mechanisms of Arc regulation by PKC. First, serines 84 and 90 were identified as the major PKC phosphorylation sites. These residues are located in the N-terminal region of the second α-helix in Arc, helix H2, which comprises amino acids 84–132. Helix H2 has already been recognized as a key locus of Arc regulation. We previously reported that H2 contains a palmitoylation motif (residues 94–98) [22], and others have identified a SUMOylation site (Lys110) [43] and binding sites for endophilin (residues 91–100) [8] and presenilin-1 (residues 91–130) [44]. H2 also contains the principal determinants of Arc self-association and mRNA binding [41]. In this study, we have focused on the potential roles of PKC phosphorylation on Arc palmitoylation and its interaction with nucleic acids.

Dynamic protein palmitoylation is a key mechanism in the control of synaptic plasticity [45,46]. In addition to targeting peripheral proteins, such as Arc, to membranes and membrane subdomains, palmitoylation can influence protein stability, subcellular localization, and protein–protein interactions. We reported a role for Arc palmitoylation in synaptic depression [22], but it is likely that additional functions of this modification will be identified. Reciprocal relationships between protein phosphorylation in the nervous system have been reported. Palmitoylation of synapsin 1, which controls synaptic vesicle clustering, is inhibited by PKA-mediated phosphorylation [47]. Likewise, palmitoylation of the dopamine transporter (DAT), which increases dopamine transport and reduces DAT downregulation, is inhibited by PKC-mediated phosphorylation [48]. In these examples, the phosphorylation sites are far apart in the protein primary structures. The close proximity of the PKC and palmitoylation sites in Arc raises the possibility that phosphorylation suppresses the direct interaction of Arc with its membrane-associated palmitoyltransferase(s). 

A long-standing puzzle concerning Arc regulation of synaptic plasticity is its ability to drive both LTP and LTD, as well as the more recently discovered activity of intercellular mRNA transport in secreted virus-like particles. It seems likely that mechanisms underlying the switching of Arc between one activity and another involve changes in its post-translational modifications and macromolecular interactions. For example, Bramham’s group has shown that SUMOylated Arc interacts with the actin regulatory protein drebrin, which has been implicated in LTP, whereas non-SUMOylated Arc interacts with elements of the AMPA receptor endocytic machinery, which contributes to LTD [22]. Our data suggest that PKC phosphorylation of Arc may inhibit some aspects of synaptic depression by suppression of its palmitoylation, as well as intercellular communication by suppression of its mRNA binding and destabilization of capsid-like Arc oligomers. In this context, it is important to note that CaMKII-mediated phosphorylation of Ser260 within the Arc Gag-like domain inhibits metabotropic receptor-dependent LTD, and that a phosphomimetic S260D mutation inhibits the formation of high-order Arc oligomers [26]. 

## 4. Materials and Methods

### 4.1. Materials

Resins: Ni^2+^-NTA resin (Roche, Indianapolis, IN, USA); Pierce glutathione agarose (Thermo Fisher Scientific, Waltham, MA, USA); Q-Sepharose (Sigma-Aldrich, St. Louis, MO, USA). Phosphorylation reagents: [γ^32^P]ATP (PerkinElmer, Waltham, MA, USA); phosphatidylserine (PS) and 1,2-dioleoyl-SN-glycerol (DAG) (Avanti Polar Lipids, Alabaster, AL, USA); protein kinase Cα (PKCα) (ProSpec, Rehovot, Israel). Proteases: papain (Sigma-Aldrich); His_6_-MBP-tagged tobacco etch virus (TEV) protease (Elizabeth Goldsmith, UT Southwestern). Mouse Arc cDNA, cloning reagents, and reagents for mutagenesis were from Thermo Fisher Scientific. Primers were from Invitrogen. pGST-parallel1 vector was a gift from Hong Zhang, UT Southwestern. The anti-Flag antibody (F1804 monoclonal anti-mouse) was from Sigma-Aldrich. Reagents for electrophoresis were from Bio-Rad Laboratories, Hercules, CA, USA. HEPES, Tris, protease inhibitor cocktail (cOmplate CO-RO), lysozyme, glutaraldehyde, Triton X-100, and other reagents were from Sigma-Aldrich. 

### 4.2. Constructs for Expression of Recombinant Wild-Type (WT) and Mutant Arc

To obtain His_6_-Arc, mouse Arc cDNA was cloned into the pQE-80L expression vector as described previously [37]. To obtain GST-Arc with a TEV cleavage site (ENLYFQ) at the N terminus, mouse Arc cDNA was cloned into the pGST-parallel1 vector. This GST-Arc construct was also used as a template for making deletion and truncation constructs by PCR. GST-Arc with a second TEV site was generated by first introducing an EcoR1 site between residues 151 and 152 and then inserting an oligonucleotide duplex of the TEV cleavage site into the EcoR1 site. Point mutations were required to correct a VG-to-EF mutation caused by the introduction of the EcoR1 site. 

### 4.3. Protein Purification

Arc and Arc deletion/truncation mutants were expressed in *E. coli* Rosetta 2 cells and purified as previously described [37]. Cells were harvested after growing for 20 h at 16 °C. His_6_-Arc or GST-Arc was extracted from the bacterial pellet with solution A (20 mM HEPES, pH 8.0, 100 mM NaCl, 5 mM DTT, protease inhibitor cocktail, and 0.2 mM phenylmethylsulfonyl fluoride (PMSF)) and 0.05 mg/mL lysozyme. The extract was centrifuged at 100,000× *g* for 1 h and the supernatant was incubated either with Ni^2+^-NTA resin (for His_6_-Arc) or glutathione resin (for GST-Arc) for at least 4 h at 4 °C. Details of His_6_-Arc preparation were described previously [37]. Extracts containing GST-Arc were incubated with glutathione resin either directly or after precipitation of Arc with ammonium sulfate at 35% saturation. The resin was first washed with solution A, then with solution A containing 0.2 % Triton X-100, and finally with solution A containing 2 M NaCl (without detergent). The washed resin was eluted either with glutathione to obtain GST-Arc or incubated with TEV protease to release Arc. Arc was dialyzed against solution B (20 mM HEPES, pH 7.5, 100 mM NaCl, 0.5 mM TCEP (reducing agent), and PMSF). If further purification was necessary, samples were subjected to anion-exchange chromatography on Q-Sepharose. His_6_-tagged TEV was removed by incubation of Arc samples with Ni^2+^-NTA. Purified Arc was aliquoted and stored at −70 °C. To remove aggregates that formed during storage, proteins were centrifuged at 230,000× *g* for 15 min at 4 °C immediately before each experiment.

### 4.4. Phosphorylation of Arc 

Arc (5 μM) was incubated for different times with ERK2 or PKC in a buffer containing 20 mM HEPES, pH 7.5, 0.1 M NaCl, 15 mM MgCl_2,_ and 0.1 mM ATP at 30 °C. To activate PKC, reaction mixtures contained additionally 1 mM CaCl_2_ and phosphatidylserine/diacylglycerol (PS/DAG) liposomes. To determine the extent of phosphorylation, [γ-^32^P] ATP was used (100 cpm/pmol). The reaction was terminated by boiling in the SDS mix and running on SDS gels. Arc bands were excised from Coomassie blue-stained gels and the amount of ^32^P was measured by scintillation counting.

### 4.5. Size-Exclusion Chromatography (SEC)

Gel filtration chromatography of Arc and Arc fragments was carried out by FPLC on a HiLoad 16/600 Superdex 200 column (GE Healthcare, Chicago, IL, USA). Precleared Arc samples in solution B were injected into the column, which was equilibrated and eluted with the same buffer. Elution patterns were monitored by absorbance at 280 nm (A_280_). A_260_ was also measured, and fractions were analyzed by SDS–PAGE and Coomassie blue staining. Calibration standards included Blue Dextran 2000 (void volume (V_o_)), thyroglobulin (R_S_: 19 nm), ferritin (R_S_: 6.1 nm), catalase (R_S_: 5.2 nm), aldolase (R_S_: 4.8 nm), BSA (R_S_: 3.6 nm), ovalbumin (R_S_: 3 nm), cytochrome c (R_S_: 1.6 nm), and ATP (total volume (V_t_)).

### 4.6. Chemical Cross-Linking 

Protein samples in 100 µL buffer B were treated at room temperature with 5 µL of freshly diluted 25% glutaraldehyde solution to obtain final glutaraldehyde concentrations from 0.0025% to 0.05%. Reactions were terminated with 10 µL of 1 M Tris-HCl, pH 8.0. Cross-linked proteins were analyzed by SDS-PAGE.

### 4.7. Analytical Ultracentrifugation (AUC)

All AUC experiments were conducted in the sedimentation velocity (SV) mode using a Beckman Coulter Optima XL-I centrifuge. Protein samples (in solution B) were prepared and incubated overnight at 4 °C. Protein solutions were introduced into the sample sectors of standard dual-sectored charcoal-filled Epon centerpieces that had been sandwiched between two sapphire windows in a cylindrical housing [49]. The reference sectors were filled with a matched buffer. The assembled centrifugation cells were inserted into an An50-Ti rotor and allowed to equilibrate at the experimental temperature (20 °C) for ~2.5 h. Centrifugation was performed at 50,000 rpm, and the time-dependent protein-concentration profiles were measured using absorbance optics tuned to 280 nm. The SV data were time-stamp corrected [50] using REDATE 1.0.1 (https://www.utsouthwestern.edu/research/core-facilities/mbr/software/). The corrected data were analyzed using the *c*(*s*) methodology in SEDFIT [51,52] at a resolution of 150, and maximum-entropy regularization with a confidence level of 0.683. SEDNTERP [53] was used to calculate partial-specific volumes, solution densities, and solution viscosities. Figures were rendered in GUSSI [38].

### 4.8. Dynamic Light Scattering

DLS experiments were carried out in a Dynapro Nanostar instrument (Wyatt Technologies, Santa Barbara, CA, USA) at 25 °C in a buffer containing 20 mM HEPES pH 7.5, 100 mM NaCl, and 1 mM TCEP. Samples (5 μL at approximately 20 µM) were dispensed into a quartz cuvette and irradiated with visible light at 661 nm. Autocorrelation data were acquired by averaged ten autocorrelation functions that had been collected for 5 s each. Three technical replicates were acquired. Data were analyzed using Dynamics, which was used in the regularization mode to calculate hydrodynamic radius distributions as shown in Figure 4. Radii (R_H_) were calculated from the translational diffusion coefficient (D), directly obtained by DLS, according to the Stokes–Einstein equation: D = kT/(6πηR_H_), where k = Boltzmann’s constant, η = viscosity, and T = absolute temperature. Figures were rendered in GUSSI [38].

### 4.9. Palmitoylation

Palmitoylation was detected using the acyl-resin-assisted capture (acyl-RAC) method [40], as described in detail by Barylko et al. [22]. Briefly, HeLa cells transfected with Flag-tagged Arc, WT, or mutants, were solubilized with 2.5% SDS in 100 mM HEPES (7.5), 1 mM EDTA, 0.2 mM PMSF, protease inhibitor cocktail, and 50 mM dithiothreitol (DTT). Lysates were incubated at 40 °C for 0.5 h (to reduce potential S–S bonds) and then for an additional 4 h with methyl methanethiosulfonate (MMTS) to block free thiols. Excess MMTS was removed by protein precipitation and washing with acetone. Dried pellets were re-solubilized in 1% SDS and mixed with thiopropyl-Sepharose resin in the presence of either hydroxylamine (NH_2_OH) to cleave thioester bonds or 2 M NaCl as a control for false positives. Proteins with free thiols (i.e., from cysteines that were originally palmitoylated before NH_2_OH treatment) were captured on the resin. After extensive washing, proteins released from the resin were analyzed by SDS-PAGE and identified by immunoblotting with anti-Flag antibody in the LiCOR Odyssey system.

### 4.10. Other Methods

HeLa cells (ATCC, Manassas, VA, USA) were cultured in DMEM supplemented with 10% fetal bovine serum and antibiotics. They were transfected with Lipofectamine 2000 according to the manufacturer’s instructions and were used 20–24 h after transfection. Protein concentration was determined using the modified Lowry method [54] according to Peterson [55] with BSA as a standard. SDS-PAGE was carried out according to Laemmli [56]. For immunoblotting, proteins were transferred to nitrocellulose and immunoblotted with the indicated antibodies. Bound primary antibodies were detected and quantified using fluorescently labeled secondary antibodies in the LI-COR system.

## 5. Conclusions

We report that Arc is phosphorylated in vitro by PKC on serines 84 and 90 within the second α-helix (helix H2) in the N-terminal domain. This modification is likely to impact the cellular functions of Arc, as helix H2 contains critical determinants of Arc self-association, palmitoylation, SUMOylation, and interactions with nucleic acids. Indeed, we found that phosphomimetic mutations (Ser-to Glu) of residues 84 and 90 disrupt Arc palmitoylation, nucleic acid binding, and high-order oligomerization. Future investigations should be directed at monitoring the effects of plasticity-inducing stimuli on PKC phosphorylation of Arc in cultured neurons. To establish the functional significance of our observations, it will be necessary to introduce Arc-containing phosphomimetic (Ser-to-Glu) and phosphoinhibiting (Ser-to-Ala) mutations in neurons from Arc knockout mice.

## Figures and Tables

**Figure 1 ijms-25-00780-f001:**
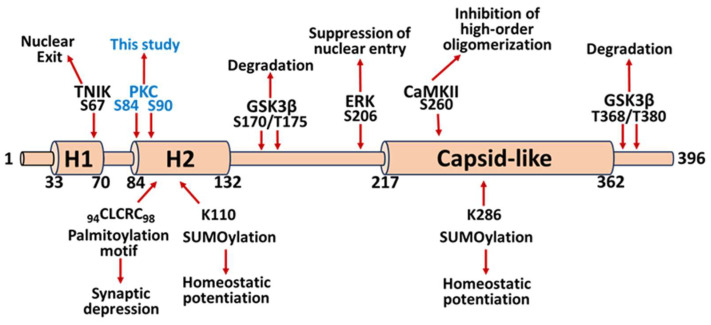
Sites of Arc post-translational modifications. Details and references are provided in the text.

**Figure 3 ijms-25-00780-f003:**
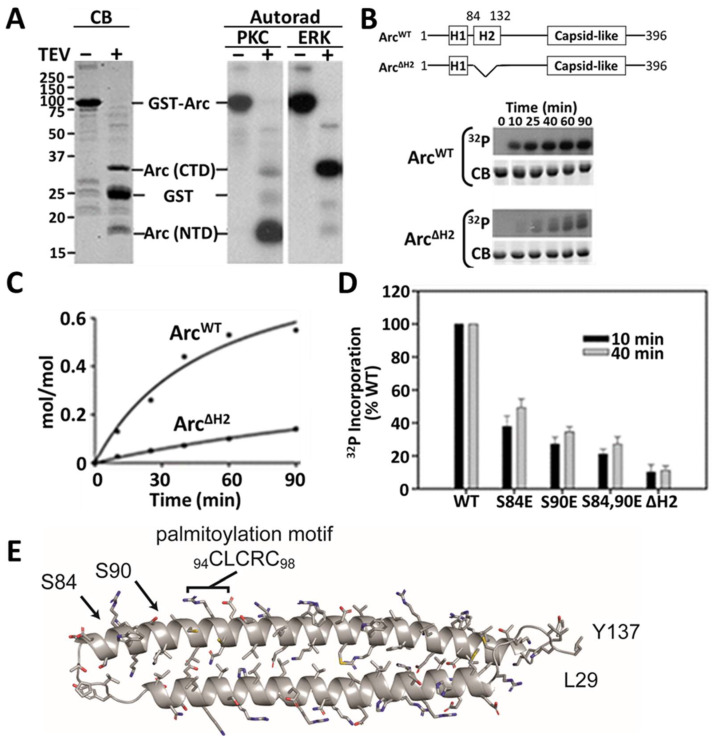
Identification of in vitro PKC phosphorylation sites in Arc. (**A**) Phosphorylation of the double-TEV GST-Arc construct, shown in Figure 2C, by PKC and ERK. Two left lanes: Coomassie blue-stained gels of proteins. Four right lanes: ^32^P incorporation into PKC- and ERK-phosphorylated GST-Arc before and after TEV proteolysis. Arc (NTD) is predominantly phosphorylated by PKC; Arc (CTD) is predominantly phosphorylated by ERK. (**B**) Scheme showing deletion of helix H2 (top panel) and time course of in vitro phosphorylation of Arc^WT^ and Arc^ΔH2^ by PKC (bottom panel). ^32^P, autoradiogram; CB, Coomassie blue-stained gels. (**C**) Quantified levels of ^32^P_i_ incorporation. (**D**) Identification of the major sites of in vitro Arc phosphorylation by PKC. Serines 84 and 90 were mutated to glutamic acid either individually or together and Arc mutants were phosphorylated with PKC for the indicated times. ^32^P_i_ incorporation was calculated as % incorporated into Arc^WT^ (estimated as 0.13, 0.15, and 0.15 mol/mol after 10 min, and 0.32, 0.34, and 0.55 mol/mol after 40 min). Phosphorylation of Arc^ΔH2^ is shown for comparison. Quantitative analysis is from three separate experiments. Arc concentration in all phosphorylation assays was 5 µM. (**E**) AlphaFold representation of Arc residues 29–137, which contains helix H1 (residues 33–70) and helix H2 (residues 84–132).

**Figure 4 ijms-25-00780-f004:**
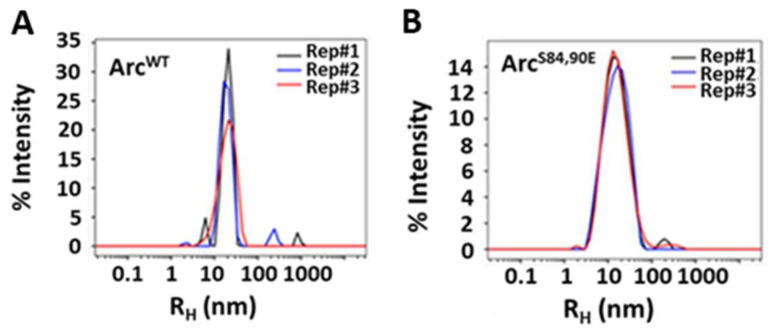
Dynamic light scattering analysis of Arc. Effect of PKC phosphomimetic mutations on Arc self-association. Three technical replicates of hydrodynamic radius distributions of ~20 μM of (**A**) Arc^WT^ and (**B**) Arc^S84,90E^ are shown. Figures were rendered in GUSSI [38].

**Figure 5 ijms-25-00780-f005:**
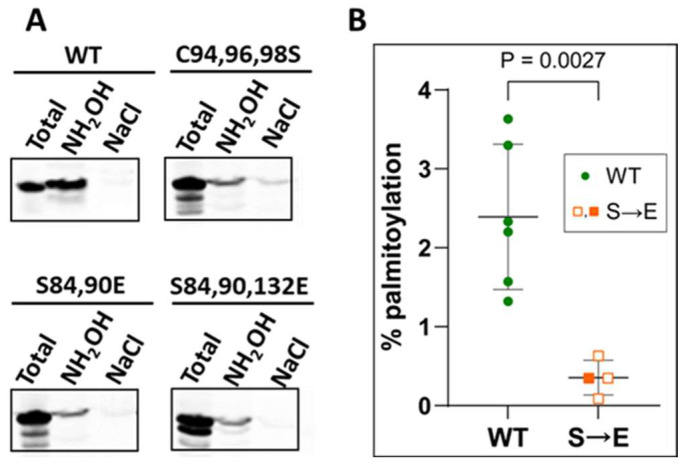
Effect of mimicking PKC phosphorylation on Arc palmitoylation. (**A**) Analysis of palmitoylation of WT and mutant Arc. Hela cells were transfected with WT Flag-Arc or its mutants, as indicated, and processed for palmitoylation as described in Section 4. C94,96,98S is a non-palmitoylated mutant, characterized in a previous report^22^. Immunoblotting was performed with anti-Flag antibodies. (**B**) Quantification of Arc palmitoylation. Filled orange square represents palmitoylation of the S84,90E mutant; open orange squares represent palmitoylation of the S84,90,132E mutant.

**Figure 6 ijms-25-00780-f006:**
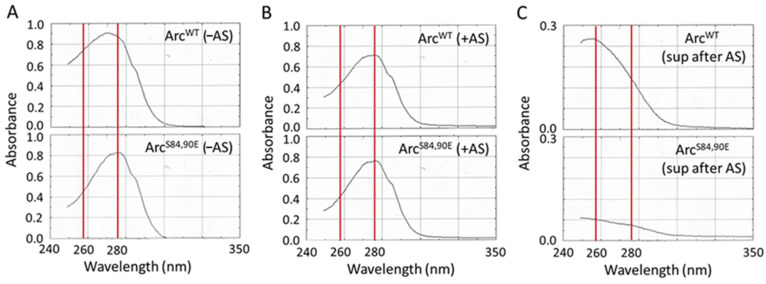
Effect of mimicking PKC phosphorylation on the interaction of Arc with nucleic acids. (**A**,**B**) UV absorbance scans of Arc^WT^ and Arc^S84,90E^ prepared without (**A**) or with (**B**) 35% ammonium sulfate precipitation. (**C**) UV absorbance scans of supernatants obtained after precipitation of Arc^WT^ or Arc^S84,90E^ with 35% ammonium sulfate. Red lines designate 260 nm and 280 nm wavelengths.

**Figure 7 ijms-25-00780-f007:**
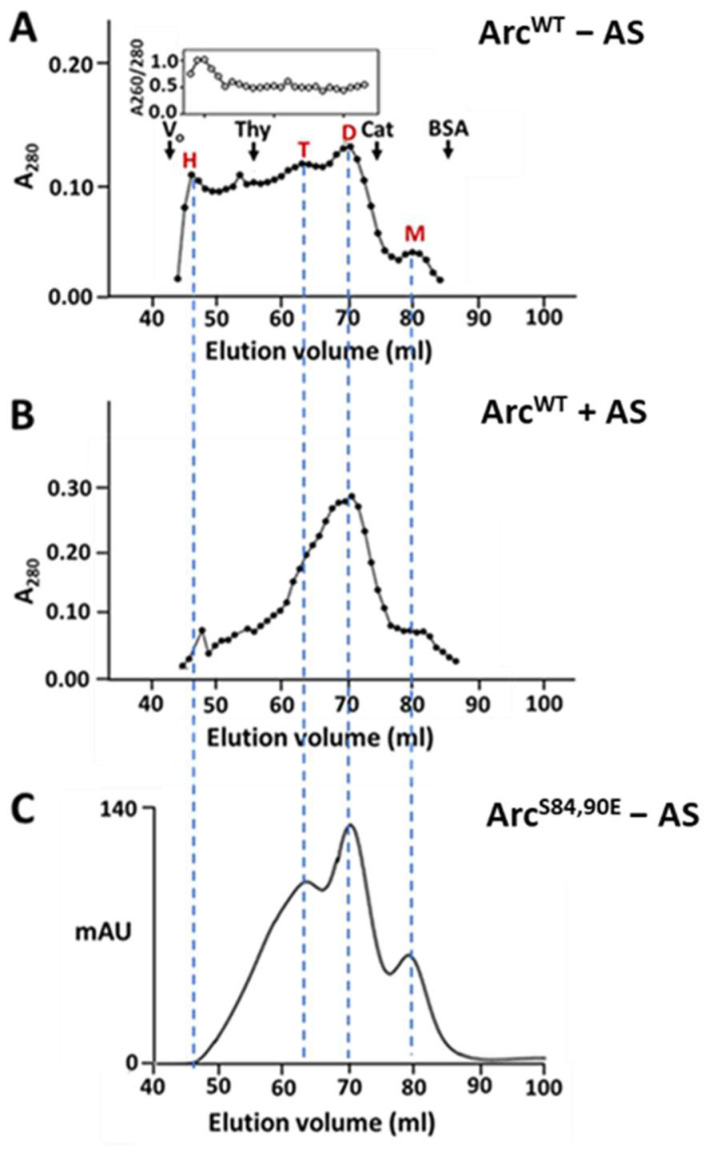
Effect of phosphomimetic mutations on the behavior of Arc during size-exclusion chromatography. (**A**,**B**) Superdex 200 elution profiles of Arc^WT^ prepared without (**A**) or with (**B**) ammonium sulfate (AS) treatment. One mL of 60 µM (−AS) or 80 µM (+AS) Arc was loaded. The inset in the upper panel shows the profile of absorbance ratios measured at 260 nm and 280 nm. Arrows above the profile designate standards: thyroglobulin (R_S_: 19 nm), catalase (R_S_: 5.2 nm), and BSA (R_S_: 3.67 nm). Red letters above the peaks indicate predicted elution positions of Arc monomers (M), dimers (D), trimers or tetramers (T), and high-order oligomers (H). (**C**) Elution profile of Arc^S84,90E^ (56 µM load) not treated with AS. All chromatography runs were carried out at 4 °C.

**Figure 8 ijms-25-00780-f008:**
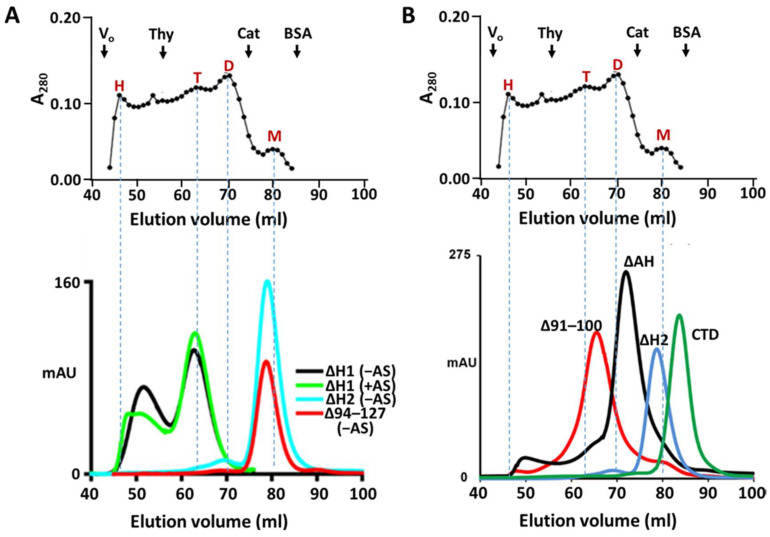
Effect of disrupting segment H2 on Arc oligomerization. (**A**) Chromatography of Arc^WT^ and Arc deletion mutants on Superdex 200. Top panel: elution profile of Arc^WT^, prepared in the absence of AS, reproduced for comparison from Figure 7A. Bottom panel: Chromatography of Arc deletion mutants on Superdex 200. Arc^ΔH1^ (25 µM) was prepared with or without AS treatment. Arc^ΔH2^ (55 µM) and Arc^94–127^ (35 µM) were prepared without AS treatment. (**B**) Chromatography of Arc mutants containing deletions within helix H2. All samples were prepared without AS treatment. Loading concentrations were as follows: Arc^Δ91–100^, 60 µM; Arc^ΔAH^ (deletion of residues 108–124), 70 µM; Arc^ΔH2^ (reproduced from panel (**A**)), 55 µM; CTD: 85 µM. Chromatography was carried out at 4 °C.

**Table 1 ijms-25-00780-t001:** Effect of mutations on nucleic acid binding.

Construct	Absorbance (260 nm/280 nm)	
	−AS	+AS
Full-length	0.86	0.55
ΔH1 (aa 33–70 deleted)	0.85	0.77
ΔH2 (aa 84–132 deleted)	0.50	0.53
Δ91–100	0.55	0.53
ΔAH (aa 108–124 deleted)	0.46	0.45
S84,90E	0.54	0.53
CTD (aa 152–396)	0.58	0.54

## Data Availability

Data are contained within the article.

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
