# Peer review of "Mimicking Protein Kinase C Phosphorylation Inhibits Arc/Arg3.1 Palmitoylation and Its Interaction with Nucleic Acids"

_ijms, 2024, doi:10.3390/ijms25020780_

Round 1
Reviewer 1 Report
Comments and Suggestions for Authors
In the present study the authors have investigated possible mechanisms by which phosphorylation by PKC regulates Arc protein. They found that serines 84 and 90 were the major PKC phosphorylation sites. These residues are found in the amino acid sequences 84–132 that make up the N-terminal portion of Arc's second α-helix, helix H2. Using phosphomimetics at these locations, they showed that PKC phosphorylation of Arc protein influences its elution in a gel filtration column, palmitoylation, and its the nucleic acid binding site. All things considered, the research suggests that PKC's phosphorylation of Arc protein at these locations influences synaptic plasticity, which is thought to be controlled by Arc. It may also inhibit interneuronal mRNA trafficking. The study is complete. The authors just need to work properly in the explanation of their rationale, findings and its plausible effect on Arc function in terms of regulation of synaptic platicity. My concerns are given below.
1. The author in the last para graph of introduction should clearly indicate the implication of PKC-phosphorylated residues to the second α-helical segment (helix H2) in the N-terminal domain of Arc. How this post translational modification is modulating Arc function in terms of LTD
2. The author has directly jumped into the palmotylation part. The author should mention clearly why they have chosen to investigate palmotylation and how changes in palmotylation could affect Arc function. The author should write this part clearly in the last paragraph of introduction.
3. What are phosphomimetic mutation. How can it affect the oligomerization of the Arc protein. The author should discuss this clearly in corresponding result section.
4. The authors have shown in Figure 4B that percent of palmotylation is reduced when S to E mutation is introduced in Arc protein as compared to the wild type. But author have not clearly explained this result in results section. Also, what is the implication of reduction in palmotylation. The authors should indicate or explain.
5. Lastly, they can justify their results that post translational modifications can modulate a protein function by citing references. They can also cite a paper where authors have shown that neutrophil survival in humans and mice is negatively impacted by mitochondrial-derived ROS and NOX2 mediated catalase S-glutathionylation (Nagarkoti et al.,2019) in the journal Inflammation
Author Response
Concern 1: The author in the last paragraph of introduction should clearly indicate the implication of PKC-phosphorylated residues to the second α-helical segment (helix H2) in the N-terminal domain of Arc. How this post translational modification is modulating Arc function in terms of LTD.
Response: The last sentence of the Introduction now states the predicted effects of PKC phosphorylation explicitly.
Concern 2: The author has directly jumped into the palmotylation part. The author should mention clearly why they have chosen to investigate palmotylation and how changes in palmotylation could affect Arc function. The author should write this part clearly in the last paragraph of introduction.
Response: We have now described the effects of all the post-translational modifications mentioned in the Introduction, including ubiquitylation, acetylation, SUMOylation, and palmitoylation. This was a major oversight on our part in the original submission, and we thank the reviewer for picking it up.
Concern 3: What are phosphomimetic mutation. How can it affect the oligomerization of the Arc protein. The author should discuss this clearly in corresponding result section.
Response: We have now defined the term “phosphomimetic” in the Introduction as follows: “In this study, we have identified serines 84 and 90 as the major in vitro PKC-phosphorylated residues. These residues reside within the second α-helical segment (helix H2) in the N-terminal domain of Arc. To mimic the effect of phosphorylation, we mutated the two serines to negatively charged glutamic acid. These “phosphomimetic” mutations resulted in a reduced propensity of Arc to undergo palmitoylation, reduced binding to nucleic acids, and reduced stability of high-order oligomers”.
Concern 4: The authors have shown in Figure 4B that percent of palmotylation is reduced when S to E mutation is introduced in Arc protein as compared to the wild type. But author have not clearly explained this result in results section. Also, what is the implication of reduction in palmotylation. The authors should indicate or explain.
Response: We have now stated that “because the phosphomimetic mutations decrease palmitoylation to essentially the same levels as mutation of the palmitoylated cysteines, we conclude that PKC phosphorylation abrogates Arc palmitoylation”. We have also now stated in the introduction the implication of reducing palmitoylation by introducing mutations that mimic PKC phosphorylation (i.e., that palmitoylation-dependent aspects of LTD are suppressed).
Concern 5: Lastly, they can justify their results that post translational modifications can modulate a protein function by citing references. They can also cite a paper where authors have shown that neutrophil survival in humans and mice is negatively impacted by mitochondrial-derived ROS and NOX2 mediated catalase S-glutathionylation (Nagarkoti et al.,2019) in the journal Inflammation.
Response: The introduction includes a citation to a review article describing how post-translational modifications modulate Arc activity: “Considering the diverse neuronal functions governed by Arc, it is not surprising that its activities are regulated by numerous posttranslational modifications [18]”. We apologize for having chosen not to cite the paper suggested by the reviewer, as Arc is apparently not subject to S-glutathionylation.
Reviewer 2 Report
Comments and Suggestions for Authors
The manuscript titled “Mimicking protein kinase C phosphorylation inhibits Arc/Arg3.1 palmitoylation and its interaction with nucleic acids” by Barylko, B.; et al. is a scientific work where the authors study the underlying regulatory mechanisms of activity-regulated cytoskeleton-associated (Arc) protein though phosphorylation post-translational modifications (specially devoted in Ser84 and Ser90). The authors employed many complementary techniques to gather this knowledge. I will recommend the present scientific manuscript for further publication in the International Journal of Molecular Sciences once all the below described suggestions will be properly fixed. There exists some points that must be covered in order to improve the scientific quality of the manuscript paper:
1) KEYWORDS. (OPTIONAL) The authors should consider to add the terms “post-translational modifications” and “neuronal activity” in the keyword list.
2) INTRODUCTION. “Activity-dependent cytoskeletal-associated protein (Arc) (…) synaptic plasticity (…) long-term spatial, taste and fear memories” (lines 48-53). The authors should also point out the pivotal role of Arc regulation in the development of neurodegenerative malignancies as Alzheimer’s disease. Recent research was devoted in this field using cultured rat hippocampal neurons as a model [1].
[1] Leung, H-W.; et al. Arc Regulates Transcription of Genes for Plasticity, Excitability and Alzheimer’s disease. Biomedicines 2022, 10, 1946. https://doi.org/10.3390/biomedicines.1008.1946.
3) “Arc is expressed predominantly (…) The latter two characteristics were also displayed the deletion of H2 and of segments within H2” (lines 53-79). Here, the authors furnish valuable insights about the factors which regulate Arc through post-translational modifications. The manuscript will benefit if a schematic representation to depict these action mechanisms is added by the authors. This could aid to the potential readers to better understand all the reaction steps involved in this process.
4) RESULTS. “Three major fragments of approximately 33, 29.5, and 24 kDa (…)” (lines 89-90). Please, the authors should homogenize the significant figures. This point should be taken into account for the rest of the main manuscript body text.
5) “Based on these results, (…) GST-tagged N-terminal (…) fragments” (lines 92-93). Please, the authors should state the full-name of all the abbreviation terms the first time they appear in the text. In this context, “glutathione S-transferase” should be defined. Then, the abbreviation should be placed between brackets. This comment should be considered for the rest of the main manuscript body text (e.g. “TEV protease sites” in line 94) .
6) “Arc152-396 has a calculated molecular weight of ~28 kDa, but an apparent molecular weight by SDS gel electrophoresis of ~28 kDa (…) protein assymmetry and/or extensive intrinsic disorder and flexibility” (lines 116-123). The authors provide the right explanation about the differences observed in terms of the expected MW of the Arc C-terminal domain which are based on its lack of globular conformation morphology. This question could be answer with crystallography data according to the native Arc protein and the respective C-terminal domain. Is this information available in literature? In case affirmative, the authors should provide a brief statement in this regard.
7) “To test whether introduction of negative charges (…) the hydrodynamic radii of ArcWT and Arc584,90E (…) using dynamic light scattering (…) hydrodynamic radii of ~22 nm and 17.5 nm, respectively” (lines 167-171). Are the data obtained by DLS accurate? The main limitation of DLS is the assumption of the ideal globular morphology of the examined sample particles which is not the case. What is the expected associated error in these measurements? Some explanation should be provided about this point.
8) “Parallel samples (…) NH2OH (…) false positives” (lines 194-195). Please, the stoichiometry number related to the chemical phormula should be indicated in subscript.
9) “2.3. Effect of phosphomimetic mutations on palmiitoylation” (lines 179-203). Did the authors observe any aggregation effect during the overexpression of the mutant proteins. In case affirmative, this effect could negatively impact in the subsequent binding with mRNA?
10) DISCUSSION. This section perfectly debates about the regulatory action mechanisms of Arc protein through post-translational modifications. Only it may be opportune to remark the importance of specific interbiomolecular interactions [2] with special focus on protein-protein and protein-mRNA to precisely control these mechanisms: “Dynamic protein palmitoylation is a key mechanism (…) subcellular localization, and protein-protein interactions” (lines 310-313).
[2] Lostao, A.; et al. Recent advances in sensing the inter-biomolecular interactions at the nanoscale – A comprehensive review of AFM-based force spectroscopy. Int. J. Biol. Macromol. 2023, 238, 124089. https://doi.org/10.1016/j.ijbiomac.2023.124089.
11) MATERIALS & METHODS. This section is clearly explained which enables the possibility to mimic the approaches employed by the authors in other labs. No actions are requested from the authors.
12) CONCLUSIONS (OPTIONAL). Even if it is not mandatory, the authors should consider to add a final “Conclusions” section to show the most relevant outcomes found in this work and briefly discuss about the potential future action lines to pursue this research highlighting the impact of the gathered knowledge in the society. Finally, the references are in the proper format of the International Journal of Molecular Sciences (No actions are requested from the authors).
Comments on the Quality of English LanguageThe manuscript is generally well-written. The authors should recheck the English in order to polish final details susceptible to be improved.
Author Response
Concern 1: KEYWORDS. (OPTIONAL) The authors should consider to add the terms “post-translational modifications” and “neuronal activity” in the keyword list.
Response: Done.
Concern 2: INTRODUCTION. “Activity-dependent cytoskeletal-associated protein (Arc) (…) synaptic plasticity (…) long-term spatial, taste and fear memories” (lines 48-53). The authors should also point out the pivotal role of Arc regulation in the development of neurodegenerative malignancies as Alzheimer’s disease. Recent research was devoted in this field using cultured rat hippocampal neurons as a model [1].
[1] Leung, H-W.; et al. Arc Regulates Transcription of Genes for Plasticity, Excitability and Alzheimer’s disease. Biomedicines 2022, 10, 1946. https://doi.org/10.3390/biomedicines.1008.1946.
Response: We have added a sentence and two references in the Introduction: “The critical role of Arc in synaptic plasticity is highlighted by findings of abnormal Arc expression and function in neurological disorders such as Alzheimer’s Disease, Parkinson’s Disease, epilepsy, and schizophrenia [3,4]”. In addition to the suggested reference to Leung et al., we have also cited a review by Chen et al. that includes a nice discussion of Arc’s connection to various diseases.
Concern 3: “Arc is expressed predominantly (…) The latter two characteristics were also displayed the deletion of H2 and of segments within H2” (lines 53-79). Here, the authors furnish valuable insights about the factors which regulate Arc through post-translational modifications. The manuscript will benefit if a schematic representation to depict these action mechanisms is added by the authors. This could aid to the potential readers to better understand all the reaction steps involved in this process.
Response: We have included a new Figure 1, which is a scheme showing almost all the known post-translational modification sites in Arc and their functions. For clarity, the several ubiquitylation and acetylation sites were not included in the diagram.
Concern 4: RESULTS. “Three major fragments of approximately 33, 29.5, and 24 kDa (…)” (lines 89-90). Please, the authors should homogenize the significant figures. This point should be taken into account for the rest of the main manuscript body text.
Response: Done.
Concern 5: “Based on these results, (…) GST-tagged N-terminal (…) fragments” (lines 92-93). Please, the authors should state the full-name of all the abbreviation terms the first time they appear in the text. In this context, “glutathione S-transferase” should be defined. Then, the abbreviation should be placed between brackets. This comment should be considered for the rest of the main manuscript body text (e.g. “TEV protease sites” in line 94).
Response: Done.
Concern 6: “Arc152-396 has a calculated molecular weight of ~28 kDa, but an apparent molecular weight by SDS gel electrophoresis of ~28 kDa (…) protein assymmetry and/or extensive intrinsic disorder and flexibility” (lines 116-123). The authors provide the right explanation about the differences observed in terms of the expected MW of the Arc C-terminal domain which are based on its lack of globular conformation morphology. This question could be answer with crystallography data according to the native Arc protein and the respective C-terminal domain. Is this information available in literature? In case affirmative, the authors should provide a brief statement in this regard.
Response: Unfortunately, there is no published crystal structure of intact Arc. The only available structure is of the so-called capsid domain (residues 217-362). Although this domain is roughly globular, our fragment comprises residues 152-396, and the non-capsid portions of this fragment (152-216 and 363-396) are intrinsically disordered. Therefore, we could not even use AlphaFold to obtain an estimate of the shape. Intrinsically disordered polypeptides often have frictional ratios of about 1.5-1.6, instead of the typical frictional ratio of less than 1.2 for hydrated globular proteins. We believe that the contribution from the disordered regions of 152-396 accounts for its apparent asymmetry detected by sedimentation velocity measurements.
Concern 7: “To test whether introduction of negative charges (…) the hydrodynamic radii of ArcWT and Arc584,90E (…) using dynamic light scattering (…) hydrodynamic radii of ~22 nm and 17.5 nm, respectively” (lines 167-171). Are the data obtained by DLS accurate? The main limitation of DLS is the assumption of the ideal globular morphology of the examined sample particles which is not the case. What is the expected associated error in these measurements? Some explanation should be provided about this point.
Response: We have addressed this issue in two ways. First, we have added the following statement to the Materials and Methods section on DLS: “Radii (RH) were calculated from the translational diffusion coefficient (D), directly obtained by DLS, according to the Stokes-Einstein equation: D = kT/(6πηRH), where k = Boltzmann’s constant, η = viscosity, and T = absolute temperature”. This statement makes clear that there is no assumption regarding the shape of the particle when estimating the radius by DLS. However, as noted by the reviewer, it is inappropriate to estimate mass (or oligomeric state) based on DLS-derived radii, as this calculation would require knowledge of the particle shape. Unfortunately, DLS values of mass are often inappropriately presented in the literature because most DLS instruments provide them upon request, and these are based on assumption of spherical shapes. To highlight this concern, we have added the following statement to the text: “We cannot use these RH values to estimate precise oligomeric states, as Arc is ~40% intrinsically disordered and, hence, is unlikely to assume an ideal globular morphology”.
Concern 8: “Parallel samples (…) NH2OH (…) false positives” (lines 194-195). Please, the stoichiometry number related to the chemical phormula should be indicated in subscript.
Response: Done.
Concern 9: “2.3. Effect of phosphomimetic mutations on palmiitoylation” (lines 179-203). Did the authors observe any aggregation effect during the overexpression of the mutant proteins. In case affirmative, this effect could negatively impact in the subsequent binding with mRNA?
Response: We did not detect aggregation after introducing these mutations, even after centrifugation at 230,000 x g for 15 minutes.
Concern 10: DISCUSSION. This section perfectly debates about the regulatory action mechanisms of Arc protein through post-translational modifications. Only it may be opportune to remark the importance of specific interbiomolecular interactions [2] with special focus on protein-protein and protein-mRNA to precisely control these mechanisms: “Dynamic protein palmitoylation is a key mechanism (…) subcellular localization, and protein-protein interactions” (lines 310-313).
[2] Lostao, A.; et al. Recent advances in sensing the inter-biomolecular interactions at the nanoscale – A comprehensive review of AFM-based force spectroscopy. Int. J. Biol. Macromol. 2023, 238, 124089. https://doi.org/10.1016/j.ijbiomac.2023.124089.
Response: The reference is very interesting, but we had difficulty applying it to our specific findings.
Concern 11: CONCLUSIONS (OPTIONAL). Even if it is not mandatory, the authors should consider to add a final “Conclusions” section to show the most relevant outcomes found in this work and briefly discuss about the potential future action lines to pursue this research highlighting the impact of the gathered knowledge in the society. Finally, the references are in the proper format of the International Journal of Molecular Sciences (No actions are requested from the authors).
Response: We have included a Conclusion section after Materials and Methods. Here we summarize our findings, their implications, and suggestions for future research.
Concern 12: The manuscript is generally well-written. The authors should recheck the English in order to polish final details susceptible to be improved.
Response: We have rechecked the English and have made some changes, which are not highlighted in yellow.